# The Presence of Bacteremia Indicates Higher Inflammatory Response and Augments Disease Severity in Adult Patients with Urinary Tract Infections

**DOI:** 10.3390/jcm11144223

**Published:** 2022-07-21

**Authors:** Chien-Chin Hsu, Pei-Chen Lin, Kuo-Tai Chen

**Affiliations:** 1Emergency Department, Chi-Mei Medical Center, Tainan 710, Taiwan; ozisun.tw@yahoo.com.tw; 2Department of Biotechnology, Southern Tainan University of Technology, Tainan 71005, Taiwan; 3Graduate Institute of Biomedical Informatics, College of Medicine Science and Technology, Taipei Medical University, Taipei 106, Taiwan; junor63@yahoo.com.tw; 4Emergency Department, Taoyuan General Hospital, Ministry of Health and Welfare, Taoyuan 330, Taiwan

**Keywords:** bacteremia, urinary tract infection, intensive care, sepsis, systemic inflammatory response syndrome (SIRS), sequential organ failure assessment (SOFA)

## Abstract

Background: This study investigated the association between the presence of bacteremia and increase in the requirement for intensive care in adult patients with urinary tract infection (UTI). The study also analyzed the differences in clinical features between patients with versus without bacteremia. Methods: We conducted a retrospective screening of the medical records of adult patients admitted during a 4-month period at a single medical center. We excluded patients with concomitant infections and patients whose urine and blood samples were not collected in the emergency department (ED). The included patients were allocated to two groups—bacteremia and nonbacteremia groups—according to the blood culture results for samples collected in the ED. Results: The study cohort comprised 637 patients, including 158 (24.8%) patients in the bacteremia group and 479 (75.2%) patients in the nonbacteremia group. Compared with the patients in the nonbacteremia group, those in the bacteremia group satisfied more systemic inflammatory response syndrome (SIRS) criteria; they had a higher white cell count, C-reactive protein level, and sequential organ failure assessment (SOFA) scores; and had a greater requirement for intensive care (bacteremia vs. nonbacteremia; SIRS: 79.1% vs. 49.9%, *p* = 0.000; leukocytosis: 68.2% vs. 57.6%, *p* = 0.000; elevation of CRP: 96.2% vs. 78.6%, *p* = 0.000; SOFA: 39.2% vs. 23.2%, *p* = 0.000; requirement for intensive care: 13.9% vs. 4.4%, *p* = 0.000, respectively). According to the results of multivariate logistic regression, bacteremia and sepsis were independent factors associated with the requirement for intensive care. Conclusions: Bacteremia increased the requirement for intensive care in patients with UTI. Physicians can identify bacteremia using inflammatory markers, the SIRS criteria, and SOFA scores.

## 1. Introduction

Urinary tract infection (UTI) is a highly common bacterial infection worldwide, and clinical manifestations range from uncomplicated cystitis to septic shock [1,2,3]. Approximately 15% to 42% of hospitalized patients with UTI exhibit bacteremia [4,5,6]. The effects of bacteremia among patients with UTI have differed among studies. A retrospective study revealed that the bacteremia is positively associated with disease severity [6]. Another observational study conducted in Israel demonstrated that bacteremia did not influence the outcomes in elderly patients with UTI [7]. Results of other studies have indicated that only patients with specific factors were at higher prognostic risk of bacteremia-complicated UTI [8,9]. Velasco et al. suggested that bacteremia is not associated with UTI in women because their blood culture results rarely indicated the need for treatment alterations [10].

Several of these conflicting results were reported by studies that compared the outcome of mortality between patients with UTI with versus without bacteremia. However, for hospitalized patients with UTI, the mortality rates are within 1.6% to 4.8% [6,9]. Due to limited case numbers and low mortality rate, the differences in mortality between UTI patients with versus without bacteremia may be statistically nonsignificant. Therefore, an appropriate outcome indicator should be selected to differentiate the effects of bacteremia in hospitalized patients with UTI.

Enterobacteriaceae, especially *E. coli*, are predominant pathogens in hospitalized patient with UTI [11]. However, the pathogens in patients with genitourinary (GU) instrumentation demonstrate a unique microbiological presentation [3,12]. Different pathogens may alter the clinical manifestations of bacteremia in UTI patients with and without GU instrumentation.

This study investigated the association of bacteremia with the increase in the requirement for intensive care in adult patients with UTI. Furthermore, the differences in clinical features and microbiological results between patients with versus without bacteremia were analyzed. We sought to discover features that can differentiate patients with bacteremia from those without bacteremia.

## 2. Methods

### 2.1. Patient Selection and Sample Collection

In this retrospective study, we screened the medical records of adult (aged ≥ 18 years) patients admitted at a single center during a 4-month period (from 1 July to 31 October 2016). Every patient with a discharge diagnosis of UTI and a urine culture performed in the emergency department (ED) was eligible for inclusion. After the charts of all eligible patients were reviewed, those satisfying all the following criteria were considered to have a diagnosis of UTI.

(1)Symptomatic criteria: Any one of the following urinary symptoms were present: dysuria, frequency, urgency, hematuria, flank pain, back pain, suprapubic pain, costovertebral tenderness on physical examination, and nonspecific symptoms. These nonspecific symptoms included nausea, vomiting, loss of appetite, shortness of breath, abdominal tenderness, or mental or functional deterioration while no other obvious infection was discovered [1,3].(2)Laboratory criteria: Urine specimens taken in ED were positive for at least two of the following four factors: pyuria (>10 leukocytes per high-power field in the sediment of a centrifuged urine specimen), bacteriuria, leukocyte esterase, nitrites [1,11].(3)Microbiological criteria: The microbiological results of the urine culture performed in the ED were positive. A positive result was defined as an organism grown in a quantitative urine culture with a concentration of >10^5^ colony-forming units/mL [11].

We excluded the patients if the following criteria were met.

(1)Concomitant infectious diseases were observed during hospitalization.(2)The patient did not undergo blood culture in the ED.(3)The microbiological results of blood culture and urine culture of the patient were discordant.(4)The urine culture or blood culture were contaminated, and the isolated pathogens included coagulase-negative staphylococci, *Bacillus* species, *Corynebacterium* species, micrococci, and *Propionibacterium* species [13].

The remaining patients were included in the study cohort. Patients with positive blood culture results were allocated to the bacteremia group and those with negative results were allocated to the nonbacteremia group.

We collected information of the patients on demographic characteristics (age and sex), immunocompromised comorbidities (liver cirrhosis, diabetes mellitus, end-stage renal disease, autoimmune disease, long-term corticosteroid use, malignancy, post-transplantation state) [6], GU instrumentations (foley catheter, percutaneous nephrostomy, and suprapubic cystostomy), prehospital state (bedridden or mobile), vital signs in the ED (arterial pressure, heart rate, temperature, respiratory rate, Glasgow Coma Scale), laboratory tests, and hospital course (presence of acute kidney injury, requirements for artificial ventilation, pressor, and intensive care, hospital length of stay, and mortality). Using the collected variables, we determined the systemic inflammatory response syndrome (SIRS) criteria and sequential organ failure assessment (SOFA) scores of the patients [14,15]. Since few patients with UTI underwent arterial blood gas analysis in the ED, we did not consider the respiratory system parameter (PaO_2_/FiO_2_) in the SOFA.

### 2.2. Comparison of Outcomes and Clinical Features between the Groups

The requirement for intensive care was the primary outcome parameter. Secondary outcomes included requirements for pressor and artificial ventilation, the presence of acute kidney injury, the length of hospital stay, and mortality. We compared the clinical features, SIRS criteria, and SOFA scores of bacteremia and nonbacteremia groups to determine the differences between the groups. Finally, we conducted a logistic regression for the patients with bacteremia and used four key variables (sepsis, SIRS, age > 64 years, and immunocompromised comorbidity) and the variable of prehospitalization state (bedridden or mobile) to determine which variable can be used to predict the requirement of the patients for intensive care [6,7,16,17].

### 2.3. Statistical Analysis and Study Approval

Statistical analyses were performed using SPSS 15 (SPSS, Chicago, IL, USA). Categorical data were expressed as frequency (%), and continuous data were expressed as mean ± standard deviation or median (in the 25% to 75% interquartile range) to reflect the data distribution. We employed a chi-square test to evaluate the differences in dichotomous variables between groups. Continuous variables were compared between groups using a Student’s *t* test and Mann–Whitney U test. In all cases, a *p* value of <0.05 was used as the threshold for statistical significance. Univariate logistic regression was performed for five variables, and multivariate analysis was conducted in statistically significant variables.

This study was reviewed and approved by the Institutional Review Board of Human Research, Chi-Mei Medical Center (IRB code: 11101-012); they waived the requirement for informed consent because the data were anonymized and collected retrospectively. We reported our findings based on the Strengthening the Reporting of Observational Studies in Epidemiology (STROBE) reporting guidelines [18].

## 3. Results

### 3.1. Patient Selection and General Description

During the 4-month study period, 8275 adult patients were admitted to the hospital through the ED, and 1050 patients discharged with a diagnosis of UTI. Among the patients with UTI, we excluded 90 patients whose blood culture was not performed in the ED, 18 patients whose urine culture in the ED had negative results, 103 patients who had concomitant infectious diseases during hospitalization, 108 patients whose blood culture and urine culture results were discordant, and 94 patients whose blood cultures or urine cultures were contaminated. The remaining 637 patients, including 158 (24.8%) patients in the bacteremia group and 479 (75.2%) patients in the nonbacteremia group, constituted the study cohort. The flow of the patient selection process is illustrated in Figure 1.

The average age of the included patients was 66.6 ± 17.9, and 63.9% patients were aged >64 years. More than two-thirds of the included patients were female (68.3%), and a large number of patients had immunocompromised comorbidities (61.5%) and were in a bedridden state (12.6%). In the ED, 364 (57.1%) patients satisfied two or more SIRS criteria, and 173 (27.2%) satisfied the criteria for sepsis (SOFA score ≥ 2). During hospitalization, 156 (24.4%) patients developed acute kidney injury, 11 (1.7%) patients required vasopressor treatment, 9 (1.4%) patients required artificial ventilation, 43 (6.8%) patients required intensive care, and 21 (3.3%) patients died.

### 3.2. Comparison between the Bacteremia and Nonbacteremia Groups

Table 1 displays the clinical features, SIRS criteria, and SOFA scores of the bacteremia and nonbacteremia groups for comparison. Regarding the demographic characteristics and prehospital state, except for a few patients in the bacteremia group who exhibited immunocompromised comorbidities and genitourinary instrumentations, the differences between the groups were nonsignificant (immunocompromised comorbidities: 51.9% vs. 64.7%, *p* = 0.004; genitourinary instrumentations: 15.8% vs. 26.3%, *p* = 0.007). Regarding the SIRS criteria, we discovered that more patients in the bacteremia group satisfied the criteria than those in the nonbacteremia group. Regarding the SOFA scores, the patients in the bacteremia group exhibited higher incidences of sepsis, thrombocytopenia, and mean arterial pressure <70 mmHg (sepsis: 39.2% vs. 23.2%, *p* = 0.000; thrombocytopenia: 34.2% vs. 2.8%, *p* = 0.000; mean arterial pressure: 9.5% vs. 2.5%, *p* = 0.000). For two common clinical useful inflammatory markers—white cell count and C-reactive protein (CRP)—the patients in the bacteremia group exhibited higher rates of leukocytosis (white cell count >10,000 cell/mL) and elevation of CRP (at >5 mg/dL) (leukocytosis: 68.2% vs. 57.6%, *p* = 0.000; elevation of CRP: 96.2% vs. 78.6%, *p* = 0.000).

Regarding the hospital course and outcome, more patients in the bacteremia group required pressor and intensive care than those in the nonbacteremia group (pressor: 4.4% vs. 0.8%, *p* = 0.003; intensive care: 13.9% vs. 4.4%, *p* = 0.000). No significant differences were noted in the incidence of acute kidney injury, requirement for artificial ventilation, and mortality (Figure 2). Hospital stays of bacteremia and nonbacteremia groups were similar (8 (7–11) vs. 7 (5–11) days, *p* = 0.161).

### 3.3. Factors Associated with the Requirement for Intensive Care

Among the five prognostic factors, bacteremia, sepsis, and SIRS criteria were significantly associated with the requirement for intensive care in the univariate analysis. A further multivariate analysis revealed that only bacteremia and sepsis were independent factors associated with the requirement for intensive care in adult patients with UTI (Table 2).

### 3.4. Results of Microbiological Studies

In the bacteremia group, all blood culture results agreed with the urine culture results. Therefore, we used the blood culture results for microbiological studies. Gram-negative bacilli constituted the majority of the pathogens (bacteremia vs. nonbacteremia; 98.7% vs. 86.9%). Fungus constituted 1.3% pathogens in the bacteremia group, and gram-positive cocci and fungus constituted 9.1% and 4.0% pathogens in the nonbacteremia group.

In both bacteremia and nonbacteremia groups, *Escherichia coli* (bacteremic vs. nonbacteremia: 81.6% vs. 54.5%) and *Klebsiella* species (bacteremic vs. nonbacteremia: 5.1% vs. 10.0%) were the two most common pathogens, followed by *Proteus* species (bacteremic vs. nonbacteremia: 4.4% vs. 5.1%), *Pseudomonas* species (bacteremic vs. nonbacteremia: 3.2% vs. 9.8%), *Enterobacter* species (bacteremic vs. nonbacteremia: 2.5% vs. 1.1%), *Citrobacter* species (bacteremic vs. nonbacteremia: 1.3% vs. 2.3%), and *Morganella* species (bacteremic vs. nonbacteremia: 0.6% vs. 1.1%). All fungal infections were caused by *Candida* species (bacteremic vs. nonbacteremia: 1.3% vs. 4.0%), whereas *Enterococcus* species and *Streptococcus* species (4.4% and 4.0%) were common isolated pathogens in the nonbacteremia group. Figure 3 showed the isolated microorganisms in the bacteremia and nonbacteremia groups.

## 4. Discussion

In this single center, retrospective study involving admitted adult UTI patients, we observed that patients with bacteremia had an increased requirement for intensive care. Elevated white cell count, CRP, SIRS criteria, and SOFA score are associated with complicated bacteremia. Therefore, for UTI patients, the presence of high inflammatory markers, SIRS criteria or SOFA score are indications for blood culture examination and infer the potential need for intensive care.

Retrospective studies from Spain and Taiwan have reported that UTI is the most common cause of bacteremia [19,20]. However, the clinical impact of bacteremia in patients with UTI has rarely been considered. In patients with nosocomial pneumonia and soft tissue infection, bacteremia increased the mortality rate [21,22]. In patients with peritoneal dialysis peritonitis, bacteremia prolonged hospital stays and did not decrease the survival rate [23]. This study’s results from the two-group comparison and multivariate logistic regression indicated that bacteremia increased the requirement for intensive care in patients with UTI. Physicians should be aware of this poor prognostic factor and should closely monitor the hemodynamic state when bacteremia is suspected in patients with UTI.

Consistent with previous reports, this study demonstrated that patients with UTI who had bacteremia fulfilled more SIRS criteria than those without bacteremia [6]. Additionally, the results of our study and several other studies have indicated that patients with bacteremia tend to have a higher white cell count and CRP level than those without bacteremia [4,24,25]. Physicians can thus use these markers to identify patients with bacteremia. Patients with UTI who satisfy the SIRS criteria with high scores or display inflammatory markers should be prescribed blood cultures to determine bacteremia-related complications.

In addition to high scores of the SIRS criteria, patients with bacteremia also demonstrated higher SOFA scores and better satisfied the definition of sepsis than patients without bacteremia. Thrombocytopenia and hypotension were the most common, prominent, and statistically significant features. Furthermore, the patients with bacteremia scored higher on the Glasgow Coma Scale <15 than those without bacteremia, and the levels of creatinine and total bilirubin were higher; however, these factors were not statistically significant. Since sepsis is associated with the increased requirement for intensive care, patients with bacteremia exhibited higher severity than those without bacteremia.

Regarding the microbiological results, gram-negative bacilli accounted for more than 86% to 98% of all pathogens, with *E. coli* ranking the highest, consistent with previous reports {4,7,9}. Additionally, other Enterobacteriaceae (*Klebsiella* species and *Proteus* species) and *Pseudomonas* species were common between the two groups. The incidence of *Pseudomonas* species infection is higher in the nonbacteremia group, which may contribute to a higher percentage of patients with genitourinary instrumentations. This finding is consistent with previous studies that patients with indwelling foley catheters are more commonly infected with *Pseudomonas* species [12,17]. The rankings of gram-negative bacilli of bacteremia and nonbacteremia groups were similar, indicating that gram-negative bacilli have resembling abilities to cause complicated blood stream infections. Although gram-positive cocci constituted 9% of the pathogens, none of the gram-positive cocci caused bacteremia. Patients in both the bacteremia and nonbacteremia groups presented with candida infections.

### Limitations

This study has several limitations. We had no standard protocol for adult patients with UTI, and some of the eligible patients were excluded from the final analysis. We also had no data on clinically important characteristics, such as the timing and effectiveness of antibiotics administration, and the reasons for the requirement for intensive care and mortality. We selected only five markers, excluding a few collected clinical features, for logistic regression analysis on the factors related to the requirement for intensive care. However, previous studies have shown that the included markers are crucial prognostic factors, two of which—SOFA score and SIRS criteria—comprise multiple vital signs, laboratory tests, and clinical features. We assumed that these markers were easy to obtain in the ED and sufficiently accounted for the clinical features of patients with UTI. Finally, the data were derived from the database of a single hospital and may not be applicable to other patient groups.

## 5. Conclusions

Bacteremia and sepsis were associated with more requirements for intensive care in patients with UTI. Additionally, patients with bacteremia scored higher in the SIRS criteria and SOFA score and exhibited higher inflammatory markers compared with patients without bacteremia. Therefore, the markers identified in this study aid in the identification of bacteremia and infer the potential need for intensive care. Gram-negative bacilli accounted for the majority of all pathogens. Enterobacteriaceae (*E. coli*, *Klebsiella* species, and *Proteus* species) and *Pseudomonas* species were common in bacteremia and nonbacteremia groups. Further prospective studies with preset protocolized managements are indicated to determine the influences of complicated bacteremia in adult patients with UTI.

## Figures and Tables

**Figure 1 jcm-11-04223-f001:**
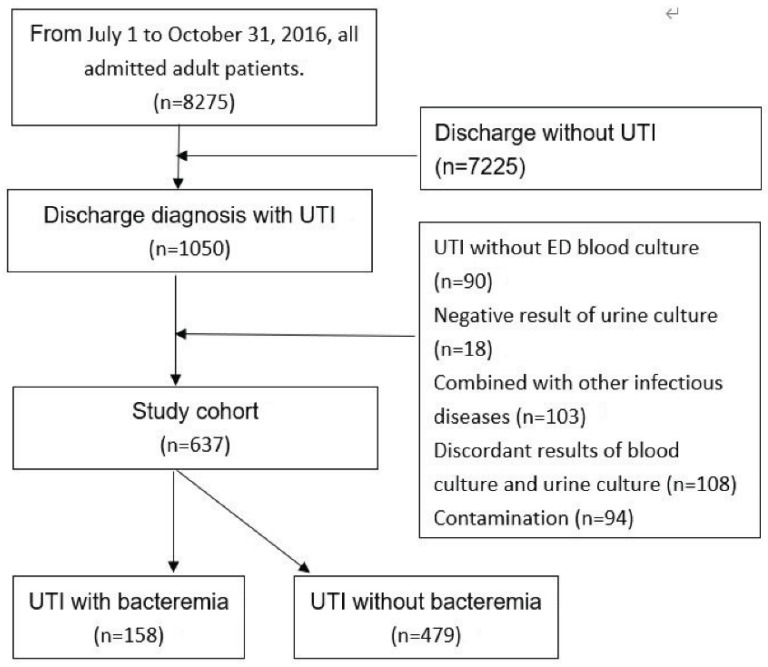
Flow of the patient selection process.

**Figure 2 jcm-11-04223-f002:**
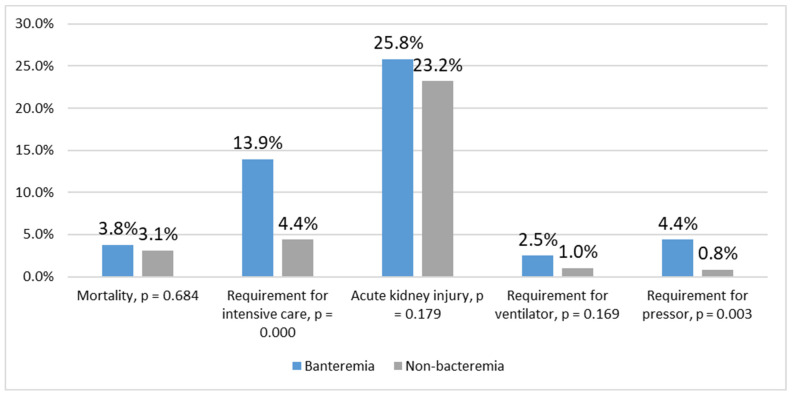
Outcome. Requirements for intensive care and a pressor were significantly greater in the bacteremia group.

**Figure 3 jcm-11-04223-f003:**
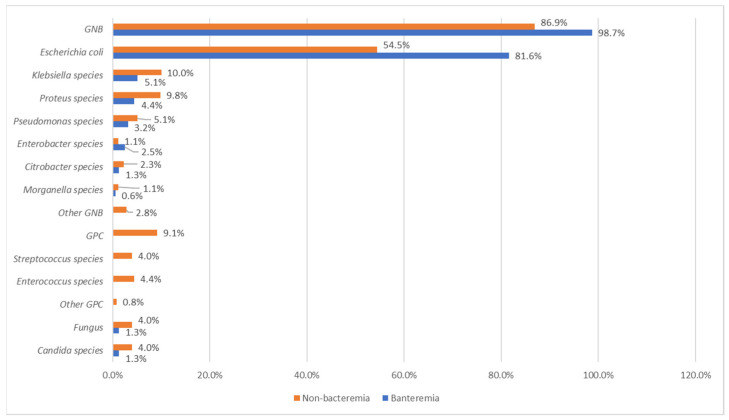
Results of microbiological studies.

**Table 1 jcm-11-04223-t001:** Comparison between the bacteremia and non-bacteremia groups.

Demographic Data	Bacteremia	Non-Bacteremia	*p*
Sex (female)	72.8%	66.8%	0.161
Age (>64 years)	55.1%	62.4%	0.101
Comorbidity	51.9%	64.7%	0.004
Bedridden	9.5%	13.6%	0.289
GU instrumentation ^†^	15.8%	26.3%	0.007
SIRS criteria ≥ 2	79.1%	49.9%	0.000
Temperature > 38, <36 °C	54.4%	28.4%	0.000
Heart rate > 90/min	77.2%	60.3%	0.000
Respiratory rate > 20/min	17.1%	9.8%	0.013
WBC > 12,000 *, <4000, band > 10%	69.5%	57.6%	0.009
Sepsis (SOFA score ≥ 2)	39.2%	23.2%	0.000
Platelet < 150,000 μL	34.2%	2.8%	0.000
Creatinine > 1.2 mg/dL	34.8%	32.8%	0.634
GCS < 15	15.8%	20.0%	0.241
MAP < 70 mmHg	9.5%	2.5%	0.000
T bilirubin > 1.2 mg/dL	27.3%	25.5%	0.878

SIRS: Systemic inflammatory response syndrome, WBC: White cell count, SOFA: Sequential Organ Failure Assessment, GCS: Glasgow Coma Scale, MAP: Mean arterial pressure, * cell/mL, ^†^ GU instrumentation: Genitourinary instrumentation, including foley catheter, percutaneous nephrostomy, and suprapubic cystostomy.

**Table 2 jcm-11-04223-t002:** Bacteremia and sepsis are two independent factors associated with the requirement for intensive care in adult patients with UTI.

Requirement for Intensive Care	Univariate Analysis			Multivariate Analysis		
	OR	95% CI	*p*	OR	95% CI	*p*
Bacteremia	0.283	0.151–0.531	0.000	0.338	0.165–0.692	0.003
Sepsis	0.239	0.127–0.450	0.000	0.291	0.150–0.564	0.000
SIRS criteria	0.498	0.250–0.984	0.045	0.594	0.282–1.253	0.171
Age > 64 years	0.727	0.376–1.405	0.343			
Comorbidity	1.679	0.845–3.336	0.139			
Bedridden	1.510	0.753–3.028	0.246			

OR: Odds ratio; CI: Confidence interval; SIRS: Systemic inflammatory response syndrome.

## Data Availability

The source codes and data presented in this study are available on reasonable request to the corresponding author (K.T.C). The raw data are not publicly available because of a data protection policy for patient data and/or patent.

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
