# Peer review of "The Presence of Bacteremia Indicates Higher Inflammatory Response and Augments Disease Severity in Adult Patients with Urinary Tract Infections"

_jcm, 2022, doi:10.3390/jcm11144223_

Round 1
Reviewer 1 Report
This is an interesting manuscript with data on UTIs that compares patients with and without concomitant bacteremia. I have some comments that can be found below:
1. A second table, providing data on microbiology could be added in order to allow the reader to better evaluate microbiological data
2. Line 224: There are some references that are presented in a different style (superscript) than the one suggested by the journal (brackets). Please double-check
3. Line 229: In this sentence, the words in red need rephrasing, as they do not match the rest of the sentence
4. English needs some revising
5. Line 246-248: The second half of the sentence sounds wrong: ‘None of the gram-negative bacilli can cause complicated blood stream infection’. Of course they can, but not in a way that infection by that microorganism leads exclusively to a very severe infection. I feel it needs to be rephrased somehow
6. Reference 1 is not in a correct format
7. Do you have any data on previous hospitalization or previous antimicrobial use of these patients?
8. Do you have any data on the antimicrobial susceptibility of the microorganisms?
Author Response
This is an interesting manuscript with data on UTIs that compares patients with and without concomitant bacteremia. I have some comments that can be found below:
- A second table, providing data on microbiology could be added in order to allow the reader to better evaluate microbiological data
Response:
In the original version of submitted manuscript, we provided a table about data on microbiology. However, the editors suggested us to delete it and we followed the instruction. In response to your valuable comment, we added a figure 3 to allow the reader to better evaluate microbiological data.
- Line 224: There are some references that are presented in a different style (superscript) than the one suggested by the journal (brackets). Please double-check
Response:
We are sorry for the mistake. We corrected it.
- Line 229: In this sentence, the words in red need rephrasing, as they do not match the rest of the sentence
Response:
We rephrased the sentence.
- English needs some revising
Response:
We corrected several structural and grammatical errors in the revised version.
- Line 246-248: The second half of the sentence sounds wrong: ‘None of the gram-negative bacilli can cause complicated blood stream infection’. Of course they can, but not in a way that infection by that microorganism leads exclusively to a very severe infection. I feel it needs to be rephrased somehow
Response:
Thank you for your remarkable comment. We rephrased it.
- Reference 1 is not in a correct format
Response:
Thank you. We revised it.
- Do you have any data on previous hospitalization or previous antimicrobial use of these patients?
Response:
This is a retrospective chart review study. Because the enrolled patients may not admit to the study hospital on previous hospitalization. Therefore, we did not record data on previous hospitalization or previous antimicrobial use of these patients.
- Do you have any data on the antimicrobial susceptibility of the microorganisms?
Response:
Because of the limitation of this retrospective study, there were no standard criteria for the use of antibiotics in patients with UTI. Many patients underwent different antibiotics in the emergency department, in ward, and in intensive care unit. It is difficult to define the antimicrobial susceptibility of the microorganisms of included patients. We are sorry that we don’t have data on this issue.
Reviewer 2 Report
Dear authors,
I believe the Introduction must be improved, there is not sufficient information regarding the bacteriemia/sepsis and the spps. that can cause it from the literature.
I liked the discussion section and I believe the methods and results are well written.
The conclussions support the results.
Please verify the English.
Overall is a good manuscript.
Author Response
Dear authors,
I believe the Introduction must be improved, there is not sufficient information regarding the bacteriemia/sepsis and the spps. that can cause it from the literature.
Response:
We revised the Introduction and conducted a more comprehensive literature review to improve this section.
I liked the discussion section and I believe the methods and results are well written.
The conclusions support the results.
Please verify the English.
Overall is a good manuscript.
Response:
We corrected several structural and grammatical errors in the revised version.
Reviewer 3 Report
- Title is confusion and need to be more informative.
- In abstract, there is repetitive sentences in the results.
- Introduction was not be enriched and purpose of this study should be revised.
- The authors should suggested their criteria for inclusion to the study with more details.
- Results of microbiological studies need to be more comparable between two groups.
- Conclusion need to be objective with sentence for further perspective.
Author Response
- Title is confusion and need to be more informative.
Response:
In response to your comment, we revised the title.
- In abstract, there is repetitive sentences in the results.
Response:
We deleted the repetitive sentences.
- Introduction was not be enriched and purpose of this study should be revised.
Response:
We revised the Introduction and the purpose of this study to improve this section.
- The authors should suggest their criteria for inclusion to the study with more details.
Response:
We revised the criteria for inclusion to the study.
- Results of microbiological studies need to be more comparable between two groups.
Response:
We added a bar graph in figure 3 to compare the causative organism distribution in bacteremia and nonbacteremia group.
- Conclusion need to be objective with sentence for further perspective.
Response:
We added a sentence in the end of Conclusion for further perspective.
Reviewer 4 Report
In this study, the authors investigated the association between the presence of bacteremia and an increase in the requirement for intensive care in adult patients with urinary tract infection (UTI). In my opinion, this study is interesting. The conclusion is supported by the results. However, I have several concerns to be addressed as follows:
- Introduction: More comprehensive literature reviews on UTI and its causative organisms, clinical manifestations, and complications, as well as bacteremia, including its definition, detection, and complications, are required.
- Methods: The IRB approval code should be provided.
- Results: Length of hospital stay was listed as a secondary outcome, but it was not compared between those with and without bacteremia in the results.
- Results: In Table 2, only significant factors in the univariate analysis should enter the multivariate analysis.
- Results: Results of microbiological studies: It is feasible to represent the causative organism distribution in bacteremia and non-bacteremia groups as a bar graph rather than numerical in the text.
- There are several structural and grammatical errors that require careful English language revision.
Author Response
In this study, the authors investigated the association between the presence of bacteremia and an increase in the requirement for intensive care in adult patients with urinary tract infection (UTI). In my opinion, this study is interesting. The conclusion is supported by the results. However, I have several concerns to be addressed as follows:
Introduction: More comprehensive literature reviews on UTI and its causative organisms, clinical manifestations, and complications, as well as bacteremia, including its definition, detection, and complications, are required.
Response:
We revised the Introduction and conducted a more comprehensive literature review to improve this section.
Methods: The IRB approval code should be provided.
Response:
We provided the IRB code in section Method, line 118.
Results: Length of hospital stay was listed as a secondary outcome, but it was not compared between those with and without bacteremia in the results.
Response:
We added the description of hospital stay in section Result, line 159.
Results: In Table 2, only significant factors in the univariate analysis should enter the multivariate analysis.
Response:
We deleted the result of insignificant factors in multivariate analysis and revised section Method, line 115-116.
Results: Results of microbiological studies: It is feasible to represent the causative organism distribution in bacteremia and non-bacteremia groups as a bar graph rather than numerical in the text.
Response:
We added a bar graph in figure 3 to represent the causative organism distribution in bacteremia and nonbacteremia group.
There are several structural and grammatical errors that require careful English language revision.
Response:
We corrected several structural and grammatical errors in the revised version.
Round 2
Reviewer 1 Report
The manuscript has been improved during the revision process
Reviewer 3 Report
well revised.
This manuscript is a resubmission of an earlier submission. The following is a list of the peer review reports and author responses from that submission.